# Growth and neurodevelopmental outcomes of preterm and low birth weight infants in rural Kenya: a cross-sectional study

Susanne P Martin-Herz [1], Phelgona Otieno,[2] Grace M Laanoi,[3,4] Vincent Moshi,[2] Geofrey Olieng'o Okoth,[2] Nicole Santos,[5] Dilys Walker[5,6]

SPM-H and PO contributed equally.

SPM-H and PO are joint first authors.

For numbered affiliations see end of article.

**Correspondence to**
Susanne P Martin-Herz;
Susanne.MartinHerz@ucsf.edu

## ABSTRACT

**Objective** Data on long-term outcomes of preterm (PT) and low birth weight (LBW) infants in countries with high rates of neonatal mortality and childhood stunting are limited, especially from community settings. The current study sought to explore growth and neurodevelopmental outcomes of PT/LBW infants from a rural community-based setting of Kenya up to 18 months adjusted age.

**Design** Cross-sectional study.

**Setting** Migori County, Kenya.

**Participants** Three hundred and eighty-two PT/LBW infants (50.2% of those identified as eligible) from a cluster randomised control trial evaluating a package of facility-based intrapartum quality of care interventions for newborn survival consented for follow-up.

**Outcome measures** Caregiver interviews and infant health, growth and neurodevelopmental assessments were completed at 6, 12 or 18 months±2 weeks. Data included sociodemographic information, medical history, growth measurements and neurodevelopmental assessment using the Ten Questions Questionnaire, Malawi Developmental Assessment Tool and Hammersmith Infant Neurological Examination. Analyses were descriptive and univariate regression models. No alterations were made to planned data collection.

**Results** The final sample included 362 PT/LBW infants, of which 56.6% were moderate to late PT infants and 64.4% were LBW. Fewer than 2% of parents identified their child as currently malnourished, but direct measurement revealed higher proportions of stunting and underweight than in national demographic and health survey reports. Overall, 22.7% of caregivers expressed concern about their child's neurodevelopmental status. Neurodevelopmental delays were identified in 8.6% of infants based on one or more standardised tools, and 1.9% showed neurological findings indicative of cerebral palsy.

**Conclusions** Malnutrition and neurodevelopmental delays are common among PT/LBW infants in this setting. Close monitoring and access to early intervention programmes are needed to help these vulnerable infants thrive.

**Trial registration number** NCT03112018.

### STRENGTHS AND LIMITATIONS OF THIS STUDY

⇒ This study used directly administered, standardised neurodevelopmental assessment tools to enhance evaluation at the community-level.

⇒ The sample included largely moderate to late preterm (PT) infants, with predominantly normal or low birth weight (LBW), as opposed to very or extremely PT/LBW infants and, therefore, may underestimate true rates of neurodevelopmental delays or disability.

⇒ The study design did not allow direct comparison to term, appropriate birth weight controls.

⇒ It was not possible to investigate factors contributing to poor growth or neurodevelopmental outcomes through multivariate analyses due to sample size constraints.

to 25%–50% of all neonatal deaths and 12% of under-5 mortality worldwide.[1 2] Additionally, close to 1 million PT survivors experience neurodevelopmental impairments each year, and PT birth is the fifth leading cause of disability adjusted life years in East Africa.[3–5] However, there is a paucity of data on the long-term outcomes of both PT and LBW infants in countries with high neonatal mortality rates (NMR), particularly from community settings.[6] In countries with an NMR>5, global estimates suggest approximately 24.6% of PT survivors are at risk of moderate or severe neurodevelopmental impairment and 32.5% of mild neurodevelopmental disability; however, these estimates are based on only seven datasets, all in settings with neonatal intensive units (NICUs).[3]

Data from community-based PT/LBW samples in areas without NICUs are extremely limited, meaning outcomes of the majority of PT/LBW infants born in low- or middle-income countries (LMIC) are not represented in current estimates.[6 7] Three community-based, rural cohort studies from Malawi,

### INTRODUCTION

Complications associated with preterm (PT) birth and low birth weight (LBW) contribute

Rwanda and Uganda exist, showing PT or LBW babies to be significantly more likely than term infants to have died between 6 weeks and 24 months adjusted age, with death rates twice as high for PT infants at 1 and 2 years than for term infants.[8–10] Survivors were more commonly wasted or underweight.[8 10] Additionally, caregivers of PT infants were significantly more likely to express concern about their child's development than caregivers of term infants; up to two-thirds of PT/LBW infants in the Rwandan sample showed developmental delays on a standardised, validated caregiver-report developmental screening tool at an average age of 22.5 months.[9 10] PT survivors were also significantly more likely to have neurodevelopmental delays on directly administered assessments than term counterparts, with particular deficits in the language and fine motor domains. Being underweight or malnourished was significantly associated with delays for both term and PT infants.[8 10]

In Kenya, an estimated 12% and 10.5% of births are PT and LBW, respectively.[11 12] In Migori County, where the current study took place, rates of malnutrition in children under-5 include stunting in 26.4%, underweight in 8.6% and wasting in 4%.[13] One study from a Kenyan urban, academic centre followed very LBW (<1500 g) infants for 2 years postdischarge and found 11.7% (95% CI 6.2 to 17.1) had cerebral palsy, 9.2% (95% CI 4.2 to 16.9) had cognitive delay and 26.7% (95% CI 12.2 to 36.9) had functional disability.[14] However, this sample is likely not representative of rural sites that lack NICU services.

Early interventions (eg, physio, occupational and speech therapies, family support and training) increasingly show improvements in long-term outcomes of PT and other at-risk babies, both in high-income settings and LMIC, highlighting the need for additional studies to better understand growth and neurodevelopment of PT/LBW infants across community settings.[6 15 16] The current study leveraged the Preterm Birth Initiative Kenya (PTBi-K) cohort[11] to explore growth and neurodevelopment of PT/LBW infants up to 18 months adjusted age in Migori County, Kenya and provides data towards better understanding of health and neurodevelopmental outcomes among PT/LBW infants at the rural, community level.

## MATERIALS AND METHODS
### Design
This cross-sectional study was conducted between October 2018 and May 2019 among a subset of mothers and babies previously enrolled in PTBi-K, a cluster randomised control trial (cRCT) of a package of interventions to improve quality of care during labour and the immediate postnatal period and evaluate the intervention's impact on stillbirth and neonatal survival. The protocol and primary results of this cRCT have been published elsewhere.[11 12] The current cross-sectional study was not designed to evaluate the impact of the cRCT intervention package.

### Setting
The current study was conducted in Migori County, Kenya. The county is mostly rural, has poor access to healthcare and has higher infant and under-5 mortality than national statistics (50 vs 39 per 1000 live births, and 82 vs 52 per 1000 live births, respectively).[13]

### Study participants and sampling strategy
Participants in the parent cRCT were identified from maternity registers. Eligible participants were LBW (<2500 g at birth) or PT (gestational age <37 weeks with birth weight <3000 g) infants delivered at one of 17 facilities across the county. A list of potentially eligible infants, alive at 28 days and approaching 6, 12 or 18 months±2 weeks of age was created, with age adjusted for PT status if the infant was born at less than 37 weeks' gestation. Recruitment was sequential toward the goal sample size across combined cRCT arms, as this follow-up study was not designed to evaluate the impact of the cRCT intervention.

A priori calculation of sample size using the Cochran's method was based on the caregiver-report Ten Questions Questionnaire (TQQ) in a community-based study of PT versus term infants in Malawi.[10 17] The calculated target sample size was n=183 per age group to detect a delay prevalence of 0.139 with a power of at least 80% and precision of 0.05.

### Procedures
Caregivers of eligible infants were contacted via phone, and a standard participation invitation script was used to explain the study. Appointments were scheduled at a study facility nearest the family's home. All consent forms and questionnaires were translated and back translated from English to Kiswahili and Dholuo.

Pregnancy, birth and neonatal course data were extracted from the cRCT database and confirmed with the caregiver when possible. Assessors were blind to the child's birth weight and gestational age, and questions regarding these variables were not asked at the study visit. The sequence of assessments was: (1) caregiver interview for sociodemographic information, medical history including growth, illness and development, and the TQQ; (2) direct neurodevelopmental assessments including the Malawi Developmental Assessment Tool (MDAT) and Hammersmith Infant Neurological Examination (HINE); and (3) physical examination including anthropometric measurements. Details of the anthropometric measurement standardised guidelines and the three neurodevelopmental assessment tools are in online supplemental tables 1 and 2.

All assessments were conducted in a conducive environment, when the child was settled and in relatively stable health, and complied with health and safety procedures. The research team consisted of clinical officers and nurses, all trained in study procedures and certified to conduct neurodevelopmental assessments. Two team members were present for each assessment, with one conducting

the assessment and one observing and recording findings. A paediatrician trained in all study procedures provided consultation and regular supervision.

After assessment, feedback on the child's neurodevelopment and health was given to the caregiver and their concerns were addressed. Caregivers were also given information on nutrition, danger signs for common childhood illnesses and simple games to play with their child. Children identified with any significant health or developmental concern, such as hearing impairment, acute malnutrition or neurodevelopmental delay, were referred to appropriate follow-up care customised to the identified need (eg, audiology, nutrition support), with costs of up to four care visits covered by the study.

Data collection was paper based, with subsequent entry into a Microsoft Access database. Double entry and verification to test for logical sequence, discrepancies and outliers was completed. Data were deidentified and stored on an encrypted server within a locked study facility. Efforts to address potential bias included sequential recruitment toward sample size goal, reporting of differences between consenting individuals and the eligible sample, similar procedures at multiple sites to reduce loss to follow-up risk that might be associated with travel to a central location, and blinding of assessors as to child's birth weight and gestational age.

### Patient and public involvement

For the larger parent study in which participants were involved, national and community advisory boards provided input on intervention priorities. Health facility providers, managers and local authorities were involved in implementation activities and influenced the focus and content of those activities based on their roles and priorities.

While caregiver participants were not involved in the design or conduct of this cross-sectional study, other than being a participant, findings specific to their child's data were shared directly with caregivers at the visit. If health conditions or neurodevelopmental delays were identified, clinical referrals were made as well.

### Statistical analysis

Analyses involved the use of descriptive statistics, as well as univariate regression models. Descriptive statistics involved the use of frequencies and proportions for categorical variables, and mean, median, range, IQR and SD for continuous variables. Sociodemographic and clinical factors associated with neurodevelopmental delay and malnutrition in infants were examined in univariate logistic regression models using the total dataset without age categorisation due to small sample size. Risk of neurodevelopmental delay or malnutrition was computed as an OR with a confidence level of 95%. All analyses were completed using STATA V.13.0 Stata/MP.

Child medical experiences were summarised as past medical illnesses (since birth) or current medical status (within 2 weeks of the assessment). MDAT and HINE total and domain scores were calculated. MDAT scores were investigated using two methods. First, the MDAT was noted as failed overall if a child was unable to complete two or more items in any one domain that would be expected to be passed by 90% of the normal reference population at their age.[18] Second, developmental z-scores were calculated using the most current MDAT Scoring Application (beta test version v1.1), and scores were dichotomised as either typical ($>-2$ SD of mean) or delayed ($\leq -2$ SD of mean). For the HINE, a score of <64 was used, as this has been shown to be 98% predictive of walking at 2 years with a sensitivity of 85% for PT children.[19] TQQ findings were described per age group, with overall caregiver concern noted if one or more items were endorsed. Apart from each assessment's categorisation of neurodevelopmental delay, a composite dichotomous neurodevelopmental delay variable was created, with a child considered to have delay if their score met delay criteria on at least one of the three neurodevelopmental tools.

For growth, WHO child growth standards were used in calculation of z-scores as provided in the STATA igrowup package.[20] Nutritional status z-scores of weight for age (WAZ), length for age (LAZ) and weight for length (WLZ) were calculated.[21] Outcomes were categorised into normal ($\geq -1$ for WAZ and LAZ; $\geq -1$ to $\leq 2$ for WLZ), at risk ($\geq -2$ to $<-1$), moderate ($<-2$ to $\geq -3$) or severe ($<-3$). Overweight and obese were defined as WLZ>2 to $\leq 3$ and WLZ>3, respectively. A composite dichotomous malnutrition variable was created with those meeting moderate or severe criteria in at least one of the three nutritional z-score variables considered malnourished.

All available data were included in the analyses. There were few missing datapoints, and any cases of missingness for pregnancy, infant and child health characteristics are noted in tables 1 and 2. No datapoints were missing for the MDAT or the HINE. One 12-month-old did not have a complete TQQ. Records with missing data were omitted only for each respective analysis.

### RESULTS

Of 761 eligible infants, 564 (74.1%) of caregivers were located. A total of 28 infants (3.7% of eligible) had died after 28 days of life and prior to study contact. While the specific causes of death for these infants are not known, a larger verbal and social autopsy study of the full parent study sample was conducted.[22] Of the 382 live babies consented for assessment (50.2% of eligible infants), six were not assessed due to acute illness at the time of appointment. The final sample included in analysis consisted of 362 infants (47.6% of eligible infants) with viable data, of which 155, 159 and 48 were 6, 12 and 18 months of age, respectively (figure 1). The target sample size of 183 per age group was not reached due to the parent study ending earlier than expected and a national health worker strike that particularly restricted the pool of eligible 18-month-old participants.

**Table 1** Delivery and immediate postnatal period characteristics

| Age at assessment | 6 months n (%) | 12 months n (%) | 18 months n (%) | All n (%) |
|---|---|---|---|---|
| **Neonatal factors** | | | | |
| Gender | | | | |
| Male | 57 (36.8) | 64 (40.3) | 23 (47.9) | 144 (39.8) |
| Female | 98 (63.2) | 95 (59.8) | 25 (52.1) | 218 (60.2) |
| Multiple pregnancy (twins) | 50 (32.3) | 46 (28.9) | 10 (20.8) | 106 (29.3) |
| Gestational age (weeks) | | | | |
| ≥37* | 59 (38.1) | 45 (28.3) | 10 (20.8) | 114 (31.5) |
| 34 to <37 | 60 (38.7) | 78 (49.1) | 24 (50.0) | 162 (44.8) |
| 32 to <34 | 16 (10.3) | 18 (11.3) | 9 (18.8) | 43 (11.9) |
| 28 to <32 | 17 (11.0) | 12 (7.6) | 5 (10.4) | 34 (9.4) |
| 22 to <28 | 3 (1.9) | 3 (1.9) | 0 | 6 (1.7) |
| Unknown | 0 | 3 (1.9) | 0 | 3 (0.8) |
| Birth weight (g) | | | | |
| 2500–2999† | 50 (32.3) | 58 (36.5) | 21 (43.8) | 129 (35.6) |
| 1500–2499 | 94 (60.7) | 97 (61.0) | 27 (56.2) | 218 (60.2) |
| 1000–1499 | 7 (4.5) | 3 (1.9) | 0 | 10 (2.8) |
| 500–999 | 4 (2.6) | 1 (0.6) | 0 | 5 (1.4) |
| Apgar—5 min | | | | |
| 0–3 | 0 | 0 | 1 (2.1) | 1 (0.3) |
| 4–6 | 7 (4.5) | 4 (2.5) | 0 | 11 (3.0) |
| ≥7 | 141 (91.0) | 144 (90.6) | 47 (97.9) | 332 (91.7) |
| Unknown | 7 (4.5) | 11 (6.9) | 0 | 18 (5.0) |
| Admitted to newborn unit (yes) | 28 (18.1) | 22 (13.8) | 8 (16.7) | 58 (16.0) |
| 'Special care' in first month (yes) | 59 (38.0) | 59 (37.1) | 11 (22.9) | 129 (35.6) |
| Oxygen | 19 (32.2) | 8 (13.6) | 3 (27.3) | 30 (23.3) |
| Phototherapy | 3 (5.1) | 4 (6.8) | 0 (0) | 7 (5.4) |
| Kangaroo mother care | 52 (88.1) | 56 (89.8) | 10 (90.9) | 118 (91.5) |
| **Maternal factors** | | | | |
| Age (years) | | | | |
| <19 | 24 (15.5) | 12 (7.6) | 5 (10.4) | 41 (11.3) |
| 19–25 | 71 (45.8) | 85 (53.5) | 27 (56.3) | 183 (50.6) |
| >25 | 60 (38.7) | 62 (39.0) | 16 (33.3) | 138 (38.1) |
| Parity | | | | |
| Primigravida | 51 (32.9) | 46 (28.9) | 10 (20.8) | 107 (29.6) |
| Multigravida | 104 (67.1) | 113 (71.1) | 38 (79.2) | 255 (70.4) |
| Delivery mode | | | | |
| Vaginal | 125 (80.7) | 144 (90.6) | 40 (83.3) | 309 (85.4) |
| Caesarean | 26 (16.8) | 14 (8.8) | 8 (16.7) | 48 (13.3) |
| Unknown | 4 (2.6) | 1 (0.1) | | 5 (13.8) |

*Infants >37 weeks' gestation were included only if birth weight was <2500 g.
†Infants 2500–2999 g were included only if gestational age was <37 weeks.

### Characteristics at delivery and immediate postnatal period

Most babies were female (60.2%) and moderate to late PT (56.6%, >32 weeks' gestation; median gestational age and range=36.3 weeks (22.0–41.7)). Of infants born PT, 66.1% were late PT (34 to <37 weeks), 17.6% were moderate PT (32 to <34 weeks), 13.9% were very PT (28

**Table 2** Child characteristics at time of visit

| Age at assessment | 6 months n (%) | 12 months n (%) | 18 months n (%) | All n (%) |
|---|---|---|---|---|
| Weight for age z-score (WAZ; underweight; valid n=343)* | | | | |
| Normal | 87 (58.4) | 68 (45.3) | 27 (61.4) | 182 (53.1) |
| At risk | 43 (28.9) | 52 (34.7) | 7 (15.9) | 102 (29.7) |
| Moderate | 13 (8.7) | 20 (13.3) | 7 (15.9) | 40 (11.7) |
| Severe | 6 (4.0) | 10 (6.7) | 3 (6.8) | 19 (5.5) |
| Length for age z-score (LAZ; stunting; valid n=351)* | | | | |
| Normal | 71 (46.7) | 62 (40.5) | 20 (43.5) | 153 (43.6) |
| At risk | 47 (30.9) | 47 (30.7) | 8 (17.4) | 102 (29.1) |
| Moderate | 24 (15.8) | 26 (17.0) | 11 (23.9) | 61 (17.4) |
| Severe | 10 (6.6) | 18 (11.8) | 7 (15.2) | 35 (10.0) |
| Weight for length z-score (WLZ; wasting; valid n=339)* | | | | |
| Normal | 107 (73.3) | 89 (59.7) | 31 (70.5) | 227 (67.0) |
| At risk | 22 (15.1) | 38 (25.5) | 7 (15.9) | 67 (19.8) |
| Moderate | 5 (3.4) | 11 (7.4) | 4 (9.1) | 20 (5.9) |
| Severe | 5 (3.4) | 7 (4.7) | 2 (4.6) | 14 (4.1) |
| Overweight | 6 (4.1) | 4 (2.7) | 0 | 10 (3.0) |
| Obese | 1 (0.7) | 0 | 0 | 1 (0.3) |
| Composite malnutrition (underweight/stunted/wasting)† | | | | |
| Normal | 111 (71.6) | 100 (62.9) | 26 (54.2) | 237 (65.5) |
| Malnourished | 44 (28.4) | 59 (37.1) | 20 (41.7) | 123 (34.0) |
| Missing | 0 | 0 | 2 (4.2) | 2 (0.6) |
| Past medical illnesses (birth until study evaluation) | | | | |
| Pneumonia | 9 (5.8) | 13 (8.2) | 6 (12.5) | 28 (7.7) |
| Diarrhoeal disease | 63 (40.7) | 107 (67.3) | 30 (62.5) | 200 (55.2) |
| Seizures | 8 (5.2) | 22 (13.8) | 4 (8.3) | 34 (9.4) |
| Malaria | 55 (35.5) | 107 (67.3) | 43 (89.6) | 205 (56.7) |
| Serious febrile illness/meningitis | 28 (58.3) | 41 (26.5) | 84 (52.8) | 153 (42.3) |
| Cough for >2 weeks | 13 (8.4) | 26 (16.4) | 5 (10.4) | 44 (12.2) |
| Malnutrition | 2 (1.3) | 3 (1.9) | 3 (6.2) | 8 (2.2) |
| Skin infections | 26 (16.8) | 51 (32.1) | 15 (31.3) | 92 (25.4) |
| Current medical illness (in past 2 weeks) | | | | |
| Acute febrile illness | 3 (1.9) | 1 (0.6) | 0 | 4 (1.1) |
| Gastroenteritis/dysentery | 21 (13.5) | 20 (12.6) | 6 (12.5) | 47 (13.0) |
| Acute malnutrition | 2 (1.3) | 2 (1.3) | 0 | 4 (1.1) |
| Respiratory tract infection/pneumonia | 33 (21.3) | 48 (30.2) | 13 (27.1) | 94 (26.0) |
| Others‡ | 17 (11.4) | 18 (11.3) | 5 (10.4) | 40 (11.0) |
| Referred for further care | 13 (8.4) | 15 (11.3) | 5 (10.4) | 33 (9.1) |

*Normal (≥−1 for WAZ and LAZ; ≥−1 to ≤2 for WLZ), at risk (≥−2 to <−1), moderate (<−2 to ≥−3), severe (<−3). Overweight WLZ>2 to ≤3, obese WLZ>3.
†Composite malnutrition includes infants who were either underweight, stunted or wasted.
‡Other illnesses included abscess (1), thrush (4), scabies (8), dermatitis (3), skin infection (18), anaemia (1), convulsions (3), otitis media (1), congenital cataract (1).

to <32) and only 2.5% were extremely PT (22 to <28). Birth weight was over 2500 g for 35.6%, and more than 90% had 5 min Apgar scores ≥7. Sixteen per cent were admitted to the newborn unit, and 35.6% needed special care (ie, oxygen, phototherapy, kangaroo mother care) in the first month of life. Approximately 50.6% of mothers

Figure 1: Study Flow Diagram

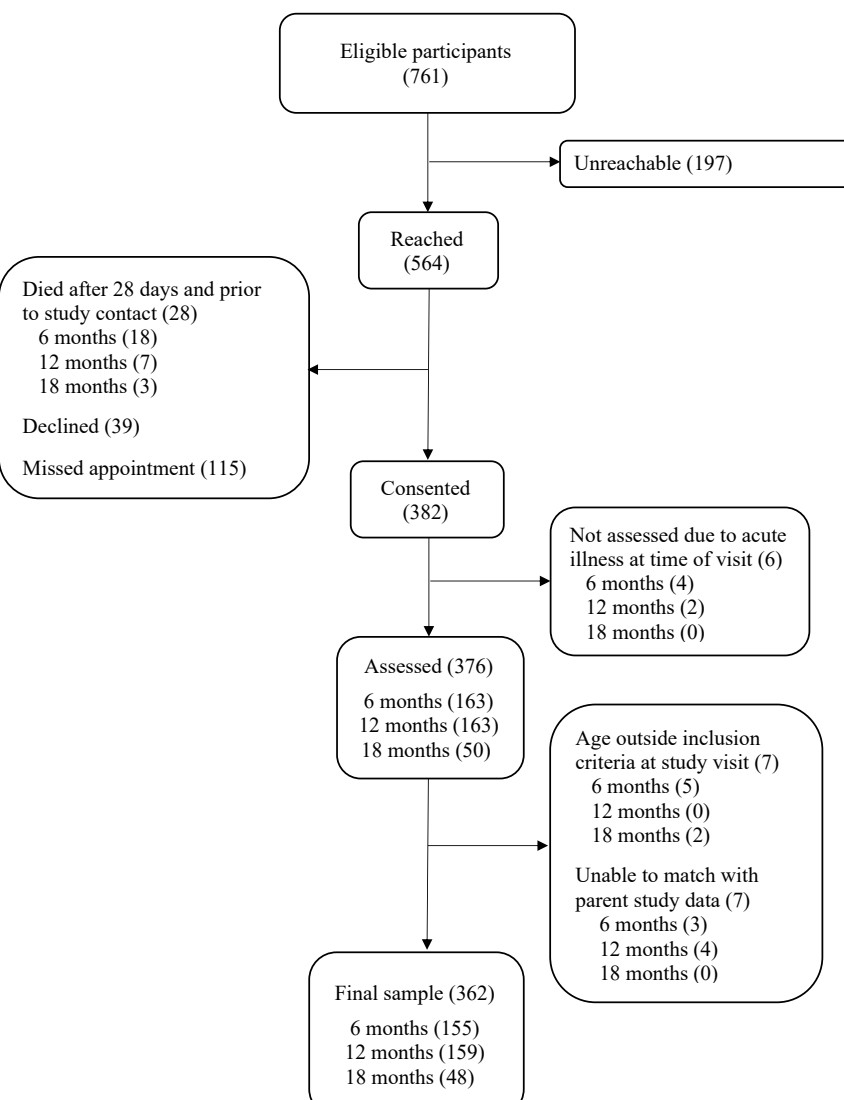

**Figure 1** Study flow diagram.

were aged 19–25. Most were multiparous (70.4%) and 13% of deliveries were by C-section (table 1).

Compared with the eligible pool of caregivers and infants from the parent study, mothers in the current study were older on average (24.7 years vs 23.6 years, t=3.16, p<0.005), and babies were more likely female (60.2% vs 52.8%, $\chi^2$=7.73, p=0.02). The two groups did not differ significantly in other key demographic variables (online supplemental table 3).

### Growth and health
Anthropometric measurement and caregiver-reported health findings are in tables 2 and 3. The prevalence of stunting, underweight and wasting in the study population were 27.4%, 17.2% and 3.3%, respectively. The proportions of children with malnutrition increased with infant age. Moderate to severe malnutrition was significantly more common in males than females (OR 2.53, 95% CI 1.62 to 3.97), and in babies born after multiple

gestation (OR 1.72, 95% CI 1.08 to 2.75) or with birth weight 1500 to 2499 g (OR 1.73, 95% CI 1.07 to 2.81).

The most common illnesses reported as ever experienced by participants included malaria (56.7%), diarrhoeal disease (55.2%) serious febrile illness (42.3%); and in the past 2 weeks prior to assessment, respiratory tract infections (26%).

### Neurodevelopment
Delays on one or more of the standardised neurodevelopmental assessment tools were identified in 8.6% of infants (tables 4 and 5). The 12-month-old infants were more likely to show delays than infants of the other two age groups, with gross motor and personal-social (MDAT z-score) areas most impacted. Seven children (1.9%) showed HINE findings indicative of cerebral palsy. In univariate analysis, a HINE score concerning for cerebral palsy was more likely in children born by C-section (OR 9.27, 95% CI 2.0 to 42.8) and was significantly associated

**Table 3** Univariate analyses for malnutrition

| Infant and maternal variables | Malnutrition (underweight) | Malnutrition (stunting) | Malnutrition (wasting) | Malnutrition (under/stunt/wast) |
|---|---|---|---|---|
| Gender | | | | |
| Male | 2.52 (1.42 to 4.48)** | 2.98 (1.83 to 4.83)*** | 1.54 (0.76 to 3.14) | 2.53 (1.62 to 3.97)*** |
| Female | 1.0 | 1.0 | 1.0 | 1.0 |
| Mother's age | | | | |
| <19 | 0.83 (0.32 to 2.15) | 0.71 (0.32 to 1.60) | 1.84 (0.67 to 5.10) | 1.00 (0.49 to 2.03) |
| 19–25 | 1.0 | 1.0 | 1.0 | 1.0 |
| >25 | 0.99 (0.55 to 1.80) | 0.89 (0.54 to 1.47) | 1.10 (0.50 to 2.40) | 0.99 (0.62 to 1.58) |
| Multiple pregnancy | | | | |
| Yes | 1.98 (1.11 to 3.53)* | 1.45 (0.88 to 2.39) | 1.48 (0.71 to 3.08) | 1.72 (1.08 to 2.75)* |
| No | 1.0 | 1.0 | 1.0 | 1.0 |
| Mode of delivery | | | | |
| Vaginal | 1.0 | 1.0 | 1.0 | 1.0 |
| Caesarean | 1.02 (0.45 to 2.32) | 0.71 (0.34 to 1.51) | 1.76 (0.72 to 4.31) | 0.95 (0.50 to 1.81) |
| Apgar1 score | | | | |
| ≤5 | 0.29 (0.09 to 0.93)* | 0.45 (0.15 to 1.34) | 0.59 (0.12 to 2.79) | 0.35 (0.12 to 1.05) |
| 6–7 | 0.76 (0.19 to 2.98) | 1.24 (0.34 to 4.50) | 0.92 (0.15 to 5.78) | 1.06 (0.29 to 3.86) |
| >7 | 1.0 | 1.0 | 1.0 | 1.0 |
| Apgar2 score | | | | |
| ≤5 | 0.20 (0.03 to 1.46) | 0.56 (0.09 to 3.44) | 0.33 (0.03 to 3.26) | 0.35 (0.06 to 2.14) |
| 6–7 | 0.44 (0.05 to 4.37) | 0.94 (0.11 to 7.73) | 0.55 (0.04 to 8.27) | 0.57 (0.07 to 4.64) |
| >7 | 1.0 | 1.0 | 1.0 | 1.0 |
| Gestational age | | | | |
| ≥37 | 1.0 | 1.0 | 1.0 | 1.0 |
| 33 to <37 | 0.90 (0.49,1.66) | 0.98 (0.59,1.64) | 0.61 (0.28,1.30) | 1.00 (0.16,1.61) |
| 28 to <33 | 0.61 (0.19,1.94) | 0.77 (0.32,1.88) | 0.43 (0.09,1.99) | 0.77 (0.34,1.77) |
| 22 to <28 | 2.97 (0.46,18.94) | PF | 4.43 (0.68,28.89) | 0.93 (0.62,5.27) |
| Birth weight | | | | |
| 2500–2999 | 1.0 | 1.0 | 1.0 | 1.0 |
| 1500–2499 | 1.38 (0.74 to 2.55) | 1.86 (1.10 to 3.18)* | 1.31 (0.60 to 2.86) | 1.73 (1.07 to 2.81)* |
| 1000–1499 | 0.65 (0.08 to 5.49) | 3.44 (0.97 to 12.21) | PF | 2.35 (0.67 to 8.21) |
| 500–999 | 1.47 (0.15 to 13.96) | 1.03 (0.11 to 9.65) | 2.65 (0.27 to 26.04) | 1.88 (0.30 to 11.74) |
| HINE | | | | |
| Normal | 1.0 | 1.0 | 1.0 | 1.0 |
| Delayed | 3.75 (0.82 to 17.22) | 1.06 (0.22 to 5.58) | 7.28 (1.56 to 34.03)* | 1.46 (0.32 to 6.61) |
| MDAT (pass/fail) | | | | |
| Normal | 1.0 | 1.0 | 1.0 | 1.0 |
| Delayed | 4.08 (1.63 to 10.19)** | 3.50 (1.46 to 8.40)** | 2.38 (0.75 to 7.59) | 3.21 (1.35 to 7.65)** |
| MDAT (z-scores) | | | | |
| Normal | 1.0 | 1.0 | 1.0 | 1.0 |
| Delayed | 6.48 (1.69 to 24.92)** | 4.18 (1.15 to 15.16)* | 4.77 (1.14 to 20.04)* | 8.07 (1.69 to 38.61)** |
| TQQ | | | | |
| Normal | 1.0 | 1.0 | 1.0 | 1.0 |
| Delay | 2.03 (1.09 to 3.78)* | 0.97 (0.55 to 1.71) | 2.24 (1.05 to 4.80)* | 1.40 (0.84 to 2.34) |

***P value <0.001.
**P value <0.01.
*P value <0.05.
PF: no variability due to low numbers causes the model to perfectly predict failure or success.
HINE, Hammersmith Infant Neurological Examination; MDAT, Malawi Developmental Assessment Tool; TQQ, Ten Questions Questionnaire.

**Table 4** Neurodevelopmental outcomes

| Age at assessment | 6 months n (%) | 12 months n (%) | 18 months n (%) | All n (%) |
|---|---|---|---|---|
| Delayed by MDAT | | | | |
| Pass/fail criteria | | | | |
| Gross motor | 6 (3.9) | 9 (5.7) | 0 | 15 (4.1) |
| Fine motor | 1 (0.7) | 2 (1.3) | 1 (2.1) | 4 (1.1) |
| Language | 0 | 2 (1.3) | 1 (2.1) | 3 (0.8) |
| Personal social | 1 (0.7) | 2 (1.3) | 1 (2.1) | 4 (1.1) |
| Total MDAT* | 8 (5.2) | 12 (7.6) | 3 (6.3) | 23 (6.4) |
| ≤−2 SD from mean | | | | |
| Gross motor | 0 | 10 (6.3) | 0 | 10 (2.8) |
| Fine motor | 6 (3.9) | 4 (2.5) | 1 (2.1) | 11 (3.0) |
| Language | 5 (3.2) | 3 (1.9) | 2 (4.2) | 10 (2.8) |
| Personal social | 3 (1.9) | 15 (9.4) | 0 | 18 (5.0) |
| Total MDAT* | 2 (1.3) | 6 (3.8) | 2 (4.2) | 10 (2.8) |
| Delayed by HINE | 5 (3.2) | 1 (0.6) | 1 (2.1) | 7 (1.9) |
| Neurodevelopmental delay† | 12 (7.7) | 15 (9.4) | 4 (8.3) | 31 (8.6) |
| Ten Questions Questionnaire | | | | |
| Total with one or more concerns | 18 (11.6) | 43 (27.0) | 21 (43.8) | 82 (22.7) |

*A fail score on the total MDAT can occur with a fail in any one or more subscales, thus this number does not represent the sum of children failing on the domain scores.
†Neurodevelopmental delay defined as a fail on one or more of the three evaluation criteria, MDAT pass/fail, MDAT z-score (≤−2 SD from mean) or HINE.
HINE, Hammersmith Infant Neurologic Examination; MDAT, Malawi Developmental Assessment Tool.

with wasting (OR 7.28, 95% CI 1.56 to 34.03). Neurodevelopmental delay was more likely in males (OR 3.55, 95% CI 1.62 to 7.79) and in infants who were underweight (OR 4.01, 95% CI 1.80 to 8.94), stunted (OR 2.96, 95% CI 1.39 to 6.33) or wasted (OR 2.76, 95% CI 1.03 to 7.36). Overall, 22.7% of caregivers expressed some concern about their child's neurodevelopment on the TQQ.

As described previously, this study recruited infants who had participated in a cRCT. The number of infants with neurodevelopmental delay were small in both control and intervention groups, and the sample was not large enough to be adequately powered to detect significant group differences if present. These data are provided for review in online supplemental tables 4 and 5, but should be interpreted with caution.

## DISCUSSION

This study describes growth and neurodevelopmental outcomes for a rural community sample of PT/LBW survivors. Infants were similar in gestational age to other community-based samples from countries with NMR>5 and constituted a relatively low-risk sample of PT/LBW infants compared with high-resource contexts or LMIC settings with available NICU care. Only 27% were born at the county's tertiary referral hospital, with the remaining born at other rural facilities. Surviving infants would thus be expected to have better outcomes than their counterparts requiring neonatal intensive care in urban settings of Africa.

Rates of stunting and underweight were higher than locally reported data, suggesting a higher risk of malnutrition in the current PT/LBW sample than in the general population of young children in the local community. Direct comparison to growth data from available community-based African samples is complicated by differences in country under-5 malnutrition rates when these studies took place.[13 23 24] Nonetheless, findings are concerning, particularly given low parental awareness (fewer than 3% expressed concern for acute or chronic malnutrition) and apparently limited detection or intervention at routine child health/immunisation visits. These findings suggest that future work focused on caregiver understanding of appropriate growth in infants born PT or LBW will be important to assuring early detection and management.

This study demonstrates that standardised assessments can be locally implemented to enhance neurodevelopmental evaluation at the community level. Directly administered, standardised neurodevelopmental assessment tools identified delay or disability in 8.6% of PT/LBW infants. This proportion is lower than global estimates from settings with high NMR but NICU care

**Table 5** Univariate analyses for neurodevelopmental delay

| Infant and maternal variables | Neurodevelopmental delay (HINE) OR (95% CI) | Neurodevelopmental delay (TQQ) OR (95% CI) | Neurodevelopmental delay (MDAT pass/fail) OR (95% CI) | Neurodevelopmental delay (MDAT z-scores) OR (95% CI) | Neurodevelopmental delay (HINE/MDAT pass/fail or z-scores) OR (95% CI) |
|---|---|---|---|---|---|
| Gender | | | | | |
| Male | 3.88 (0.74,20.30) | 0.95 (0.58 to 1.58) | 3.71 (1.49 to 9.27)** | 3.61 (0.92 to 14.20) | 3.55 (1.62,7.79)** |
| Female | 1.0 | 1.0 | 1.0 | 1.0 | 1.0 |
| Mother's age | | | | | |
| <19 | 3.08 (0.50,19.04) | 0.56 (0.23 to 1.35) | 0.73 (0.16 to 3.38) | 2.79 (0.64 to 12.20) | 1.45 (0.50,4.21) |
| 19–25 | 1.0 | 1.0 | 1.0 | 1.0 | 1.0 |
| >25 | 0.88 (0.15,5.35) | 0.64 (0.37 to 1.10) | 1.00 (0.41 to 2.46) | 0.53 (0.10 to 2.77) | 0.82 (0.36,1.86) |
| Multiple pregnancy | | | | | |
| Yes | 3.26 (0.72,14.83) | 1.15 (0.68 to 1.97) | 1.06 (0.42 to 2.66) | 1.04 (0.26 to 4.09) | 1.35 (0.62,2.92) |
| No | 1.0 | 1.0 | 1.0 | 1.0 | 1.0 |
| Mode of delivery | | | | | |
| Vaginal | 1.0 | 1.0 | 1.0 | 1.0 | 1.0 |
| Caesarean | 9.27 (2.01,42.82)** | 1.36 (0.68 to 2.71) | 1.00 (0.29 to 3.52) | 3.00 (0.75 to 12.04) | 2.03 (0.82,5.00) |
| Apgar score at 1 min | | | | | |
| ≤5 | 0.12 (0.02,0.55)** | 1.07 (0.29 to 3.96) | 0.31 (0.06 to 1.53) | 0.26 (0.03 to 2.36) | 0.45 (0.09,2.14) |
| >6 | 1.0 | 1.0 | 1.0 | 1.0 | 1.0 |
| Apgar score at 5 min | | | | | |
| ≤5 | PF | 1.20 (0.13 to 10.90) | 0.38 (0.08 to 1.82) | 0.34 (0.04 to 2.91) | 0.54 (0.11,2.54) |
| >6 | 1.0 | 1.0 | 1.0 | 1.0 | 1.0 |
| Gestational age | | | | | |
| ≥37 | 1.0 | 1.0 | 1.0 | 1.0 | 1.0 |
| 33 to <37 | 0.74 (0.16,3.35) | 1.10 (0.63 to 1.90) | 1.12 (0.44 to 2.86) | 1.12 (0.27 to 4.55) | 1.00 (0.45,2.25) |
| 28 to <33 | PF | 1.10 (0.44 to 2.72) | 0.96 (0.19 to 4.83) | 1.12 (0.11 to 11.14) | 1.01 (0.26,3.89) |
| 22 to <28 | PF | PF | PF | PF | PF |
| Birth weight | | | | | |
| 2500–2999 | 1.0 | 1.0 | 1.0 | 1.0 | 1.0 |
| 1500–2499 | 3.59 (0.43,30.20) | 1.11 (0.66 to 1.87) | 1.50 (0.57 to 3.96) | 0.89 (0.25 to 3.20) | 2.45 (0.97,6.19) |
| 1000–1499 | PF | 1.55 (0.38 to 6.37) | 5.04 (0.87 to 29.10) | PF | 5.17 (0.90,29.81) |
| 500–999 | PF | PF | PF | PF | PF |
| Underweight | | | | | |
| Normal | 1.0 | 1.0 | 1.0 | 1.0 | 1.0 |
| Abnormal | 3.75 (0.82,17.22) | 2.03 (1.09 to 3.78)* | 4.08 (1.63 to 10.19)** | 6.48 (1.69 to 24.92)** | 4.01 (1.80,8.94)** |
| Stunting | | | | | |
| Normal | 1.0 | 1.0 | 1.0 | 1.0 | 1.0 |
| Abnormal | 1.06 (0.20,5.58) | 0.97 (0.55 to 1.71) | 3.50 (1.46 to 8.40)** | 4.18 (1.15 to 15.16)* | 2.96 (1.39,6.33)** |
| Wasting | | | | | |
| Normal | 1.0 | 1.0 | 1.0 | 1.0 | 1.0 |
| Abnormal | 7.28 (1.56,34.03)* | 2.24 (1.05 to 4.80)* | 2.38 (0.75 to 7.59) | 4.77 (1.14 to 20.04)* | 2.76 (1.03 to 7.36)* |

***P value <0.001.
**P value <0.01.
*P value <0.05.
PF: no variability due to low numbers causes the model to perfectly predict failure or success.
HINE, Hammersmith Infant Neurological Examination; MDAT, Malawi Developmental Assessment Tool; TQQ, Ten Questions Questionnaire.

available, where one might anticipate higher-risk infants surviving. It is more comparable to, but still lower than that of other cited community-based studies.[3][8][10] A higher number of caregivers expressed developmental concerns, with more concern for older children, likely in part due

to the increase in observable developmental milestones/skills as children age.

The HINE was successfully used as an assessment for cerebral palsy or motor disability risk. Approximately 2% of children showed concern for being non-ambulatory by 2 years, and one additional child met clinical criteria for cerebral palsy but was not included in the sample due to acute illness at the time of visit. While these numbers are low, the percentage is not markedly different than the 3.4% of children with neonatal encephalopathy who had 'sub-optimal' HINE scores in a recent Ugandan study.[25] With global PT births estimated at 15 million annually, even these small percentages would translate to almost 1.3 million children with developmental delay or high risk for disability annually, highlighting the importance of targeted clinical follow-up and implementation of early intervention programmes for these at-risk infants in low-resource communities.[12 26]

In addition to malnutrition and neurodevelopmental risks, a high proportion of the sample were reported to have experienced acute childhood illness in their lifetime, including malaria, diarrhoeal disease and serious febrile illness. Children in the current study had higher rates of acute respiratory infection in the last 2 weeks than local averages for children under 5 years (26% vs 13%).[13] Increased rates of respiratory and severe infections have been documented for PT infants in other contexts, indicating that these major illnesses may differentially affect PT/LBW infants.[10] Although community data for the other illnesses are lacking, malaria is endemic in Migori County and a major cause of under-5 mortality (19%).[27]

Our data may underestimate true developmental delay/disability rates for PT/LBW infants for two reasons. First, participants were part of a larger cRCT evaluating the effect of an intrapartum and immediate postnatal intervention package on PT/LBW neonatal survival in which the control arm also received two of the four interventions. Post-hoc univariate analyses revealed no significant differences in growth or neurodevelopment between babies born at control versus intervention sites (online supplemental tables 4 and 5); however, these findings should be considered with caution due to the small sample size and because this cross-sectional study was not designed to evaluate the impact of the cRCT intervention on growth or neurodevelopmental outcomes.[11 12] Second, study participants were largely moderate to late PT infants with predominately normal or LBW, as opposed to very or extremely PT/LBW infants, and the vast majority had 5 min Apgar scores ≥7.[28] These findings are consistent with WHO data suggesting that half of babies born before 32 weeks in low-income countries will not survive; however, they suggest findings may be an underestimate of adverse outcomes of PT/LBW babies in LMIC more broadly.[29] Compared with all infants who survived to 28 days in the larger parent study, infants in this sample were more likely to be female and to have younger mothers at time of delivery (online supplemental table 3). Since 79% of infants who died prior to study contact were female, survival bias is an unlikely reason for this female predominance. However, in our small sample, males were more likely to be malnourished and have developmental delay, suggesting that additional longitudinal investigation into gender-related outcomes is warranted. Whether maternal age differences were due to differential survival or challenges in locating teen mothers is unknown; however, future research would ideally gather information on surviving PT infants among adolescent mothers in LMIC. Other important sample characteristics did not differ, suggesting the sample was largely representative of the PT/LBW population. In contrast, there is the possibility that these data may bias somewhat toward higher risk of health, growth and neurodevelopmental difficulties, since almost 30% of participating infants were born after twin pregnancies. Future studies with long-term follow-up of PT/LBW infants may consider including only singleton births or planning a priori for additional analyses comparing infants from singleton pregnancies with those born after twin pregnancies.

This study has several limitations. First, the study design did not allow for direct comparison to term, normal birth weight controls and it was not possible to investigate factors contributing to poor growth or neurodevelopmental outcomes through multivariate analyses. Additionally, there were too few babies in the highest-risk PT/LBW categories to separately investigate their neurodevelopmental outcomes. There were several constraints related to recruitment for this study. The parent study was not originally designed as a longitudinal follow-up past 28 days, and this meant that we did not have recurrent contact with caregivers between the infant turning 28 days and the follow-up study recruitment call, which occurred up to 17 months later. Additionally, a national health worker strike significantly reduced recruitment into the parent study during the birth months of 18-month-olds, markedly reducing the number of potentially eligible children at this age. Although only 6.9% of those contacted declined to participate, 25.9% were unreachable and 20.4% of those scheduled for an informational recruitment visit did not attend that visit, so it was not possible to describe the study to them in detail. The analysed sample consisted of just under 50% of the identified eligible sample (figure 1), suggesting possible selection bias in this subsample. Despite these limitations in sampling, our data contribute to the very limited follow-up data on outcomes in PT/LBW infants in community samples of LMIC. The experienced challenges in recruitment underscore the importance of setting up robust longitudinal cohorts to obtain high-quality data on the long-term outcomes of these vulnerable infants in LMIC to inform intervention and policy planning.

## CONCLUSION

The current study adds to very limited community-based literature on PT/LBW infants born in countries with high NMR and suggests higher than background rates of wasting and underweight, high rates of parental concern for development, and a clinically impactful number of children with neurodevelopmental delay or risk for disability. The results highlight the need for policies that support close monitoring of and early intervention for high-risk infants to assure PT/LBW infants in both rural and urban areas of LMIC are able to thrive.

### Author affiliations
¹Department of Pediatrics, University of California San Francisco, San Francisco, California, USA
²Center for Clinical Research, Kenya Medical Research Institute, Nairobi, Kenya
³Department of Paediatrics and Child Health, Maseno University, Maseno, Kenya
⁴Paediatric & Child Health, Jaramogi Oginga Odinga Teaching and Referral Hospital, Kisumu, Kenya
⁵Institute for Global Health Sciences, University of California San Francisco, San Francisco, California, USA
⁶Department of Obstetrics, Gynecology & Reproductive Sciences, University of California San Francisco, San Francisco, California, USA

**Acknowledgements** We wish to thank the caregivers and infants who generously gave their time to participate in this project.

**Contributors** SPM-H, PO, GMN, NS and DW contributed to study conceptualisation. VM, GO led data cleaning and analysis. SPM-H, PO, GMN, VM, GO, NS and DW contributed to data synthesis and interpretation. SPM-H and PO drafted and revised the manuscript, GMN, VM, GO, NS and DW provided edits and feedback. All authors contributed to manuscript revision and approved the final version of the manuscript for submission. SPM-H and PO as co-first authors accept full responsibility for the work and/or conduct of the study, had access to the data and controlled the decision to publish.

**Funding** This work was supported by the Bill & Melinda Gates Foundation under grant number OPP1107312 (East Africa Preterm Birth Initiative).

**Competing interests** None declared.

**Patient and public involvement** Patients and/or the public were involved in the design, or conduct, or reporting, or dissemination plans of this research. Refer to the Methods section for further details.

**Patient consent for publication** Not applicable.

**Ethics approval** This study involves human participants and was approved by University of California San Francisco Institutional Review Board (IRB #: 18-25555) and Scientific and Ethics Review Unit (SERU) of the Kenya Medical Research Institute (KEMRI) #KEMRI/SERU/CCR/0104/3668. Written authorisation was obtained from the Migori County Director of Health. Formal written informed consent procedures were completed in the preferred language of each caregiver.

**Provenance and peer review** Not commissioned; externally peer reviewed.

**Data availability statement** Data are available upon reasonable request.

**ORCID iD**
Susanne P Martin-Herz http://orcid.org/0000-0002-2474-3904

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
