## [Reviewer comments · BMJ Open]

ARTICLE DETAILS

TITLE (PROVISIONAL)	Growth and Neurodevelopmental Outcomes of Preterm and Low Birthweight Infants in Rural Kenya: a Cross-sectional Study
AUTHORS	Martin-Herz, Susanne P.; Otieno, Phelgona; Nalwa, Grace; Moshi, Vincent; Olieng'o Okoth, Geoffrey; Santos, Nicole; Walker, Dilys

VERSION 1 – REVIEW

REVIEWER	Thornton, Jim University of Nottingham
REVIEW RETURNED	24-May-2022

GENERAL COMMENTS	This is a puzzling study. The authors report 18 month follow up of a subset of babies from a cluster RCT of a package of facility-based quality of care interventions around the time of birth in Kenya. The original trial measured the effect of the intervention on preterm birth outcomes and was reported as showing that stillbirth and neonatal mortality among low-birthweight and preterm babies were reduced. However the present report pools the intervention and control groups and treats the trial as a cohort study. Since community based preterm child development assessments in Kenya are few and far between, the authors argue that readers will be interested. I rather doubt it.. The problem is there's no similar assessment of term babies, (or any other comparison group) and the study enrolment was pretty rough and ready - any birth before 37weeks or BW <2.5Kg. So it's impossible to judge whether the preterm babies are doing, better, worse or the same. At best the study as reported would be of minor local interest. However it could be easily rescued by reporting the groups as randomised. Follow up studies of interventions like this are few and far between. This would be well worth publishing. Even if there was no difference between the groups, that would be useful information. I note the present study only reports a subset of randomised babies.. This is a weakness. But it's equally a weakness of the study reported as a cohort. If the authors do decide to revise and report by group I would suggest that they also include the stillbirths and NND in the outcome tables so readers can judge the whole picture.
--

	If they wish they could still add a 3rd column for the whole cohort, i.e. the same data as in the present paper. My prediction is readers will be more interested in the outcomes by group!
--	---

REVIEWER	Dall'Asta, Andrea University of Parma, Medicine and Surgery
REVIEW RETURNED	12-Jun-2022

GENERAL COMMENTS	Thank you for inviting to review the manuscript entitled “Growth and Neurodevelopmental Outcomes of Preterm and Low Birthweight Infants in Rural Kenya: a cross-sectional study” for BMJ Open. Please see below my comments. Abstract: it is overall well written, but my main concern is that more information should be provided on the gestational age at delivery of the included cases. I hope you agree there is a huge difference between 24+0 and 36+6 in terms of short and long-term outcome. I would also comment on the concept of malnourishment: the problem is common in LMICs, particularly in poor areas, so how is it possible to discriminate between children who are malnourished due to prematurity/LBW and those who are malnourished due to poverty and deprivation? Introduction: - “Early interventions increasingly show improvements in long-term outcomes of PT and other at-risk babies” – please detail the type of interventions Results - Please detail the median or mean GA at delivery - Late PTB is usually from 34 weeks onwards - Please detail in the text actual numbers in the results section. - The fact that multiple gestations were included could represent a potential bias. - I recommend performing a sub-analysis of cases who had PTB AND postnatal diagnosis of SGA Discussion - Good
--

VERSION 1 – AUTHOR RESPONSE

REVIEWER #1 COMMENTS:

1. This is a puzzling study. The authors report 18 month follow up of a subset of babies from a cluster RCT of a package of facility-based quality of care interventions around the time of birth in Kenya. The original trial measured the effect of the intervention on preterm birth outcomes and was reported as showing that stillbirth and neonatal mortality among low-birthweight and preterm babies were reduced. However, the present report pools the intervention and control groups and treats the trial as a cohort study. Since community based preterm child development assessments in Kenya are few and far between, the authors argue that readers will be interested. I rather doubt it.

The problem is there's no similar assessment of term babies, (or any other comparison group) and the study enrolment was pretty rough and ready - any birth before 37weeks or BW <2.5Kg. So it's impossible to judge whether the preterm babies are doing, better, worse or the same. At best the study as reported would be of minor local interest.

However, it could be easily rescued by reporting the groups as randomised. Follow up studies of interventions like this are few and far between. This would be well worth publishing. Even if there was no difference between the groups, that would be useful information.

We appreciate the reviewer's constructive feedback and apologize that some details were not more clearly stated. We presented a total sample for two reasons. First, the intervention was not designed to impact later growth or neurodevelopmental outcomes, and we did not have a strong hypothesis or theoretical basis to link the intervention with these outcomes. Second, the follow-up sample was not adequately powered to detect significant differences between these two groups, should they exist. In addition, the number of infants with neurological delay in the final sample was very small (8.6% of full sample), rendering the power for this approach insufficient and the comparisons inappropriate. We therefore chose to present a pooled population of preterm/low birth weight (PT/LBW) infants across the two parent study intervention arms (intervention at time of birth). Further clarification has been added to the Design section of the Methods. (p.6) This limitation was also previously addressed in the Discussion section. (p. 16)

We continue to believe that dissemination of these findings is of strong value to a broader group working in global maternal, neonatal and child health, particularly in sub-Saharan Africa. As noted, there is a dearth of available information on outcomes of PT/LBW infants born in community settings of Africa. In their joint report, "Born Too Soon: The Global Action Report on Preterm Birth" the World Health Organization, Save the Children, Partnership for Maternal, Newborn and Child Health and the March of Dimes pointed out that "*Action for preterm birth will start from increased visibility and recognition of the size of the problem— deaths, **disability, later chronic disease**, parent suffering and wider economic loss*" (p. 76, bold font added). They further highlighted "*careful attention to follow up of premature babies (including extremely premature babies) and early identification of impairment*" (p. 76) as a key implementation action that was still needed.¹

We agree with the reviewer that some readers may be interested in seeing study findings by parent study intervention arm and that studies following PT/LBW infants after interventions in low resource community contexts will be important additions to the literature. Therefore, we have included supplemental tables to report the sample by parent study arm (intervention vs. control site) and referenced these tables both in the Results and Discussion sections. (p.14 and p. 16) In the latter, we have included clear language about how these results should be taken with caution given the small sample size. (p. 16)

Additionally, the reviewer rightly mentions that we did not have the privilege of comparing PT/LBW infants with healthy term babies. This is a key limitation raised in the Discussion section, where we describe that the study was nested in a larger trial, in which the control group received 2 of the 4 possible interventions and no healthy term babies were included in data collection. (p. 16) Given this limitation, we highlight how findings from this study compare to local malnutrition data and to available global neurodevelopmental data, both data from low- and middle-income country settings with NICU care available, as well as from community settings in countries with high NMR. (p. 15-16)

2. I note the present study only reports a subset of randomised babies. This is a weakness. But it's equally a weakness of the study reported as a cohort.

Thank you for these helpful comments. We agree with the reviewer that selection bias may be introduced given that we were only able to recruit a sub-set of the parent study's participants. We previously addressed this in the Results (p. 11) and in Supplemental Table 3, detailing that study participants did not differ in key demographic variables from the eligible pool of mother-infant dyads from the parent study. It was also described and discussed as a study limitation in the Discussion section. (p. 17)

We apologize if the current study sample was described as a cohort. It is a subsample of the parent study. We have corrected this language in the Strengths and Limitations (p. 3) and in the Discussion (p. 17).

3. If the authors do decide to revise and report by group, I would suggest that they also include the stillbirths and NND in the outcome tables so readers can judge the whole picture.

Thank you for this suggestion. Stillbirth and neonatal death (NND) by group are reported in the parent study publication (Reference # 12). For the current study, infants were enrolled after the neonatal period, with only infants still surviving at 28 weeks' postpartum considered eligible for recruitment (see *Study Participants and Sampling Strategy*, p. 6). For these reasons, stillbirths and neonatal deaths are not reported.

We do report infant deaths after 28 days that were identified at time of the recruitment call. (Results p. 9, Figure 1)

4. If they wish they could still add a 3rd column for the whole cohort, i.e. the same data as in the present paper. My prediction is readers will be more interested in the outcomes by group!

Thank you for this suggestion. Please refer to our answer for Comment #1 above.

REVIEWER #2 COMMENTS:

1. Abstract: it is overall well written, but my main concern is that more information should be provided on the gestational age at delivery of the included cases. I hope you agree there is a huge difference between 24+0 and 36+6 in terms of short and long-term outcome.

Thank you for this suggestion, and we do agree that there are differences in outcome across gestational age. To better describe the sample, we have included additional description of the range of gestational ages in the sample and the gestational age sub-categories of those infants born preterm. (p. 10)

We also already identified the low number of infants born very and extremely preterm in the sample as a limitation. (p. 17)

2. Abstract: I would also comment on the concept of malnourishment: the problem is common in LMICs, particularly in poor areas, so how is it possible to discriminate between children who are malnourished due to prematurity/LBW and those who are malnourished due to poverty and deprivation?

Thank you for this important question. Infants in the current sample did show higher rates of stunting and underweight than locally reported rates for children under 5 years of age, which was particularly concerning given low rates of caregiver recognition of malnutrition as a concern (see Introduction p. 5 and Discussion p. 15).

We have added high rates of childhood stunting to the abstract to emphasize it is an important consideration. (p. 3) We have also somewhat expanded our discussion of this area for clarity. (p. 15)

3. Introduction: "Early interventions increasingly show improvements in long-term outcomes of PT and other at-risk babies" – please detail the type of interventions

Thank you for this suggestion. We have included some specific examples of early interventions in the Introduction. (p. 5)

4. Results: Please detail the median or mean GA at delivery. Late PTB is usually from 34 weeks onwards

Thank you for this important comment. We have included the median and range gestational age of the cohort in the Results. (p. 10) Additionally, we have updated Table 1 to separately show data for moderate (32 - < 34 weeks) and late (34 - < 37 weeks) preterm infants. (p. 10)

5. Results: Please detail in the text actual numbers in the results section.

For language in which we used fractions (e.g., half, over one-third), we have provided actual numbers. (p.10)

6. Results: - The fact that multiple gestations were included could represent a potential bias.

The reviewer raises an important point. Multiple gestations were included in the parent study eligibility criteria, as multiplicity is considered a risk factor for preterm delivery, low birthweight and neonatal mortality,² and subsequent risk for neurodevelopment impairments due to lower gestational ages and birthweight.³ We have added the number of infants born in multiple pregnancies to Table 1 (p. 10) and described this risk of potential bias in the Discussion. (p. 17)

7. Results: I recommend performing a sub-analysis of cases who had PTB AND postnatal diagnosis of SGA

We appreciate the reviewer’s recommendation to analyze small-for-gestational age (SGA) preterm babies compared to appropriate-for-gestational age (AGA) preterm babies. SGA was not an available variable for preterm participants in our dataset, and the small overall sample size would likely also pose an additional limitation to this analysis. Given a recent meta-analysis showing minimal differences in neurodevelopment between preterm AGA and preterm SGA infants,⁴ and given the studies lack of adjusted or predictive analyses for outcomes, we respectfully suggest that development of this variable and further analyses of these differences would not substantially contribute to findings of this paper.

Adapted from Sania et al, 2019, Figure 3. Pooled estimates of association between child factors and development.

References

1. March of Dimes, PMNCH, Save the Children, WHO. Born too soon: The global action report on preterm birth. 2012.
2. Monden CWS, Smits J. Mortality among twins and singletons in sub-Saharan Africa between 1995 and 2014: a pooled analysis of data from 90 Demographic and Health Surveys in 30 countries. *The Lancet Global Health*. 2017;5(7):e673-e679. doi:10.1016/S2214-109X(17)30197-3
3. Lorenz JM. Neurodevelopmental Outcomes of Twins. *Seminars in Perinatology*. 2012;36(3):201-212. doi:10.1053/j.semperi.2012.02.005
4. Sania A, Sudfeld CR, Danaei G, et al. Early life risk factors of motor, cognitive and language development: a pooled analysis of studies from low/middle-income countries. *BMJ Open*. 2019;9(10):e026449. doi:10.1136/bmjopen-2018-026449

VERSION 2 – REVIEW

REVIEWER	Thornton, Jim University of Nottingham
REVIEW RETURNED	22-Dec-2022
GENERAL COMMENTS	Thank you. The authors have responded adequately to my comments.
REVIEWER	Colditz, Paul University of Queensland, UQCCR
REVIEW RETURNED	09-Oct-2022

GENERAL COMMENTS	This is a significant contribution to the literature of outcomes after preterm/VLBW birth in a community setting in Africa. The cohort was part of an existing and reported RCT. The study enables the identification of some targets for intervention. There are a few questions:  1. Are the participants equally from each of the arms of the RCT (which was a perinatal intervention to improve outcomes)? Depending on the results of the RCT, this may introduce bias. A short sentence would be sufficient to clarify this. 2. How was the sample size calculated? I am not a statistician, but have trouble understanding the description on page 8. What is the "effect size" (of 0.3) referred to in the calculation? What additional information could have been ascertained if the calculated sample size was achieved? 3. The 50% inclusion rate is a potential cause of bias. Although only 6.9% of those contacted declined to participate, 25.9% were unreachable, and 20.4% of those scheduled for visits, did not attend. Some further data analysis would be possible to ascertain whether the perinatal variables presented, differed in the analysed vs non-analysed groups. This may make the interpretation more robust.
--

REVIEWER	Maleta, Kenneth University of Malawi College of Medicine, School of Public Health and Family Medicine
REVIEW RETURNED	13-Oct-2022

GENERAL COMMENTS	Thanks for the opportunity to review the revised manuscript which is a definite improvement of the previous version. The major lingering concern amplified by the clarifications provided in the revised manuscript remain the quality of the study especially in terms of sampling errors and selection bias arising from the lack of specificity of the inclusion criteria adopted. These unfortunately can not be cured at this stage but significantly affect the reliability and validity of the findings. A few additional observations indicated below.  1. The revised version of the paper is a prevalence survey of growth and neurodevelopmental outcomes amongst preterm and low birthweight infants in rural Kenya. As clarified in the revision the authors surveyed infants and children from a pregnancy cohort studied in a cluster randomised trial which included 2368 28-day survivors (Supplemental table 3) from which 761 infants met the inclusion criteria for the cross-sectional study, i.e., being low birth weight and /or preterm at birth and surviving to 28 days. Of the sample approached (761) only about half (54%) consented to participation, and amongst those consenting, the final studied sample was less than half of the original eligible sample (362/761) (Figure 1). The attrition in this study makes it prone to sampling error which should be acknowledged and discussed. This is more so considering the estimated sample size of 183 per age group was not attained. 2. Please clarify why the authors adopted a uniform age +/- 2 weeks definition of age when the margin provided translates to between 2.5% to 7% of the target ages at 18 and 6 months respectively. This effectively erroneously reduced the sampling pool. The margins for age definition should have been adjusted for age. This may explain the small pool in the older age group of 18 months.
--

	3. Page 12 presents results of growth and development in table 2 and 3 both as descriptives (proportions) and inferential statistics from univariate analyses but the text does not clarify which variables underwent univariate analyses and the tables do not show any univariate analyses and statistics although some of these are cited in the text. It would have been better to present all the results in the respective tables. 4. Page 14 line 32 which describes OR for HINE score amongst cerebral palsy children is unclear and requires revision. Also clarify whether these models were conducted on the total sample or by age group and if the latter whether the relationships remained the same in the various age groups. If the former is true provide an explanation of that approach considering that the design specifically was interested in the outcome at various ages. 5. Figure 1, The presentation should be modified such that it clearly shows that 410 were consented and only 376 were assessed with attrition being attributed to acute illness (6) and death before assessment (28) and please correct so that reasons for deaths are presented in same way as those that declined or were unreachable. It is also important the flow diagram follows a consistent approach with reasons for exclusions presented by age group i.e., the illness exclusions should also be grouped by age. Please also indicate how many of each age group were consented. It may be easier from this point to have 3 subsidiary boxes from the consent box where attrition in each group and effective sample sizes are indicated by group. 6. Please comment on the fact that there was a 7.2% mortality in the 7-8 months of this study, translating to an IMR of almost a 100 which is almost three times the national average. 7. Page 18 line 12-13 please modify to clarify that this refers to those scheduled for appointment for a recruitment visit, otherwise it reads as if 20.4% of those scheduled for assessment visits post-consent missed their visit.
--	--

VERSION 2 – AUTHOR RESPONSE

Reviewer: 3

Dr. Paul Colditz, University of Queensland

Comments to the Author:

This is a significant contribution to the literature of outcomes after preterm/VLBW birth in a community setting in Africa. The cohort was part of an existing and reported RCT. The study enables the identification of some targets for intervention. There are a few questions:

1. Are the participants equally from each of the arms of the RCT (which was a perinatal intervention to improve outcomes)? Depending on the results of the RCT, this may introduce bias. A short sentence would be sufficient to clarify this.

All subjects from the main cluster randomized control trial (cRCT) were equally eligible for recruitment, making the sampling frame without stratification. Recruitment was sequential toward the goal sample size per age group and depended on availability of potential participants at each age and our ability to reach them by phone call or trace them with support of community health workers (CHVs).

The decision to not recruit equal numbers of participants from the cRCT study arms was intentional and made a priori, prior to initiation of study recruitment. This decision was based on two reasons: (1) the parent study intervention was not designed to impact longer-term growth or neurodevelopmental outcomes, and (2) the current follow-up study was not intended to evaluate the impact of the cRCT intervention. This has been further clarified in the Materials and Methods section. [pp. 6-7]

2. How was the sample size calculated? I am not a statistician, but have trouble understanding the description on page 8. What is the “effect size” (of 0.3) referred to in the calculation? What additional information could have been ascertained if the calculated sample size was achieved?

Our sincere apologies for this error in the manuscript. We have updated the text to correctly reflect the true allowable margin of error (precision) of 5% for a 95% confidence level as was correctly stated in the study protocol. [p. 7]

3. The 50% inclusion rate is a potential cause of bias. Although only 6.9% of those contacted declined to participate, 25.9% were unreachable, and 20.4% of those scheduled for visits, did not attend. Some further data analysis would be possible to ascertain whether the perinatal variables presented, differed in the analysed vs non-analysed groups. This may make the interpretation more robust.

Thank you for this recommendation. Supplemental Table 3 compares demographic variables of all eligible infants from the parent study to demographic variables from those who participated in this follow-up study. As noted in the discussion [p. 22-23], infants who survived to 28 days in the larger parent study differed from infants in the follow-up study sample in being more likely to be female and to have younger mothers at time of delivery. Survival bias was noted to be an unlikely reason for the female predominance, since 79% of infants who died prior to study contact were female. Other important perinatal characteristics such as gestational age at birth, APGAR scores, and mode of delivery, did not differ between groups, suggesting the study sample was largely representative of the parent study sample. Therefore, we believe the concern for bias resulting from the inclusion rate is minimal but cannot be fully ruled out. [see Supplemental Table 3, Results p. 13, and Discussion p. 21]

Reviewer: 4

Dr. Kenneth Maleta, University of Malawi College of Medicine

Comments to the Author:

Thanks for the opportunity to review the revised manuscript which is a definite improvement of the previous version. The major lingering concern amplified by the clarifications provided in the revised manuscript remain the quality of the study especially in terms of sampling errors and selection bias arising from the lack of specificity of the inclusion criteria adopted. These unfortunately cannot be cured at this stage but significantly affect the reliability and validity of the findings.

We appreciate the reviewer’s comments and note the concerns raised about the study’s inclusion criteria and sampling methodology. We have further addressed these limitations in the manuscript.

First, as previously noted (R#3, Q1), recruitment for this follow-up study was planned to sequentially sample from the parent study, depending on availability/reachability via phone call or CHV tracing. This a priori decision was made since the cRCT intervention was not designed to impact longer-term growth or neurodevelopmental outcomes and the current follow-up study was not intended to evaluate the impact of the cRCT intervention. This has been further detailed in the Discussion. [p. 22]

Second, we acknowledge the potential for sampling bias which is discussed in the limitations section of the Discussion [p. 22]. As noted above (R#3, Q3), Supplemental Table 3 compares demographic variables of all eligible infants from the parent study to demographic variables from all participants in this follow-up study sample and discusses the two differences noted. The majority of perinatal characteristics did not differ between groups, suggesting the sample was largely representative of the PT/LBW parent study population. While this reduces concern for selection bias, it cannot be fully ruled out. [see Supplemental Table 3 and p. 21]

A few additional observations indicated below.

1. The revised version of the paper is a prevalence survey of growth and neurodevelopmental outcomes amongst preterm and low birthweight infants in rural Kenya. As clarified in the revision the authors surveyed infants and children from a pregnancy cohort studied in a cluster randomised trial which included 2368, 28-day survivors (Supplemental table 3) from which 761 infants met the inclusion criteria for the cross-sectional study, i.e., being low birth weight and /or preterm at birth and surviving to 28 days. Of the sample approached (761) only about half (54%) consented to participation, and amongst those consenting, the final studied sample was less than half of the original eligible sample (362/761) (Figure 1). The attrition in this study makes it prone to sampling error which should be acknowledged and discussed. This is more so considering the estimated sample size of 183 per age group was not attained.

We have more carefully outlined these data in Figure 1 and the manuscript [Results, p. 11] and acknowledge the limitations of this sampling which we detail in the Discussion section. [pp. 21-22] While this attrition has implications for potential selection bias, we feel strongly that our findings will be valuable to the field, which currently has very limited follow-up data on preterm and low birth weight infants in LMIC community samples. We emphasize the importance of establishing and funding robust longitudinal cohorts to more fully understand the long-term outcomes of these vulnerable infants and their families, and to inform intervention and policy planning.

2. Please clarify why the authors adopted a uniform age +/- 2 weeks definition of age when the margin provided translates to between 2.5% to 7% of the target ages at 18 and 6 months respectively. This effectively erroneously reduced the sampling pool. The margins for age definition should have been adjusted for age. This may explain the small pool in the older age group of 18 months.

We appreciate this comment and understand we could have expanded this range, particularly for older infants. The main contributor to the reduced pool of 18-month-olds for recruitment, however, was a national health worker strike that took place during the months of birth of infants in this age

range. The parent study had markedly fewer infants recruited during the strike, reducing those eligible to be contacted at 18 months of age. Because of our strategy of identifying potentially eligible children as they approached the age of eligibility [see Study Participants and Sampling Strategy, p. 7], we did not have difficulty with children aging out of eligibility. The health worker strike was mentioned in the Results section [p. 11], and further discussion of this has been added in the Discussion limitation's section. [p.22]

3. Page 12 presents results of growth and development in table 2 and 3 both as descriptives (proportions) and inferential statistics from univariate analyses but the text does not clarify which variables underwent univariate analyses and the tables do not show any univariate analyses and statistics although some of these are cited in the text. It would have been better to present all the results in the respective tables.

We appreciate this suggestion and have clarified the findings in several ways. First, we have modified the statistical analysis section of the Methods to include description of the univariate analyses and variables considered (e.g., socio-demographic and clinical factors). [p. 9-10] In addition, we have added two new Tables (Tables 3 and 5) to the Results section showing univariate analysis output to support the in-text discussion of findings. [p. 15-Table 3, p. 17-18-Table 5]

4. Page 14 line 32 which describes OR for HINE score amongst cerebral palsy children is unclear and requires revision. Also clarify whether these models were conducted on the total sample or by age group and if the latter whether the relationships remained the same in the various age groups. If the former is true provide an explanation of that approach considering that the design specifically was interested in the outcome at various ages.

Models were constructed using the total sample, due to the small sample size and limited data variability resulting from few subjects with neurodevelopmental delays or malnutrition. Subcategorization by age group would have resulted in issues of model convergence (due to small sample size) or findings of perfect failure (0) or success (1) (due to no/low variability). We have added a univariate table (Table 5) to support the in-text discussion of these findings. [p. 17-18]

5. Figure 1, The presentation should be modified such that it clearly shows that 410 were consented and only 376 were assessed with attrition being attributed to acute illness (6) and death before assessment (28) and please correct so that reasons for deaths are presented in same way as those that declined or were unreachable. It is also important the flow diagram follows a consistent approach with reasons for exclusions presented by age group i.e., the illness exclusions should also be grouped by age. Please also indicate how many of each age group were consented. It may be easier from this point to have 3 subsidiary boxes from the consent box where attrition n each group and effective sample sizes are indicated by group.

We appreciate this request for clarification. As noted, 28 infants died after 28 days of age and prior to our contacting their caregivers for possible follow-up study enrollment, so consent was not obtained from these families. We do not have cause of death information for these infants, although a separate verbal and social autopsy study was completed for which these infants were eligible, and its findings provide further insights.¹ This reference has been added to the manuscript. [p. 11] In total, we

consented 382 infants to be assessed for all other study procedures. We have modified Figure 1 as recommended to include the age categories.

1. Olack B, Santos N, Inziani M, et al. Causes of preterm and low birth weight neonatal mortality in a rural community in Kenya: evidence from verbal and social autopsy. *BMC Pregnancy Childbirth*. 2021;21(1):536. doi:10.1186/s12884-021-04012-z

6. Please comment on the fact that there was a 7.2% mortality in the 7-8 months of this study, translating to an IMR of almost a 100 which is almost three times the national average.

Thank you for this important clarifying question. The listed 28 infant deaths occurred after 28 days of life but prior to our reaching families for enrollment in the current study. This has been clarified in the Figure, and the number of children in each age category has also been added. Because of the small sample, restricted only to preterm and low birthweight infants at 6, 12 or 18 months, we cannot robustly calculate an IMR from these data.

7. Page 18 line 12-13 please modify to clarify that this refers to those scheduled for appointment for a recruitment visit, otherwise it reads as if 20.4% of those scheduled for assessment visits post-consent missed their visit.

Thank you for this suggestion. This has been clarified as suggested. [p. 22]

Reviewer: 1

Prof. Jim Thornton, University of Nottingham

Comments to the Author:

Thank you. The authors have responded adequately to my comments.

Jim Thornton

Nottingham. 22 December 2022

Reviewer: 3

Competing interests of Reviewer: Nil

Reviewer: 4

Competing interests of Reviewer: None

Reviewer: 1

Competing interests of Reviewer: none21